# Cyber Risk in Health Facilities: A Systematic Literature Review

**Alberto Sardi** [1,*] , **Alessandro Rizzi** [1], **Enrico Sorano** [1] **and Anna Guerrieri** [2]

[1]  Department of Management, University of Turin, 10124 Turin, Italy; alessandro.rizzi@unito.it (A.R.);
    enrico.sorano@unito.it (E.S.)
[2]  Societè Hospitaliere D'assurances Mutuelles, 10129 Turin, Italy; anna.guerrieri@relyens.eu
*   Correspondence: alberto.sardi@unito.it

**Abstract:** The current world challenges include issues such as infectious disease pandemics, environmental health risks, food safety, and crime prevention. Through this article, a special emphasis is given to one of the main challenges in the healthcare sector during the COVID-19 pandemic, the cyber risk. Since the beginning of the Covid-19 pandemic, the World Health Organization has detected a dramatic increase in the number of cyber-attacks. For instance, in Italy the COVID-19 emergency has heavily affected cybersecurity; from January to April 2020, the total of attacks, accidents, and violations of privacy to the detriment of companies and individuals has doubled. Using a systematic and rigorous approach, this paper aims to analyze the literature on the cyber risk in the healthcare sector to understand the real knowledge on this topic. The findings highlight the poor attention of the scientific community on this topic, except in the United States. The literature lacks research contributions to support cyber risk management in subject areas such as Business, Management and Accounting; Social Science; and Mathematics. This research outlines the need to empirically investigate the cyber risk, giving a practical solution to health facilities.

**Keywords:** cyber risk; cyber-attack; cybersecurity; computer security; COVID-19; coronavirus; information technology risk; risk management; risk assessment; health facilities; healthcare sector; systematic literature review; insurance

## 1. Introduction

This paper presents a systematic literature review on cyber risk in the healthcare sector. The risk is defined as the combined probability of an unwanted event and its level of impact. It is described as a function of the probability that a given source of threat exerts a potential vulnerability and the consequent impact of this adverse event on the organization [1]. Cyber risk, also known as information technology risk, is the new management challenge of the third millennium; it affects the information and technology assets of organizations [2]. Scholars define cyber risk in different ways: "the risk involved with a malicious electronic event that disrupts business and monetary loss" [3,4], "the risk is an inherent part of a business and public life" [5], or "the risk failing information systems" [6]. The term "cyber" is referred to the cyberspace, an interactive domain composed of all digital networks used to store, modify, and communicate information. It includes all information systems used to support businesses, infrastructures, and services [7]. We here define cyber risk as "operational risks to information and technology assets that have consequences affecting the confidentiality, availability, or integrity of information or information systems" [8]. Numerous examples of cyber risks can be made; for instance, theft, disclosure of sensitive information, and business interruption [7]. Any device connected to the Internet is subject to cyber-attack [2].

Since the beginning of the millennium, scholars have investigated the "cyber risk". For instance, in the largest scientific database, Elsevier's Scopus, the first publication dates back to 2003; instead, when typing "cyber risk", around 180 documents are included in July 2020. Through this article, a special emphasis is given to the cyber risk is becoming a dangerous hazard during the COVID-19 pandemic. As written by the World Health Organization, the number of cyber-attacks is now more than five times than of the same period last year [9]. This growing trend has put in serious difficulty the healthcare sector, revealing a huge risk for all health processes. Although the sources of operational cybersecurity risk can be derived from the actions of people, systems and technology failures, failed internal processes, and external events, during the COVID-19 pandemic, the main cyber risks have derived from the action of people and systems and technology failures. Action of people is referred to as "action, or lack of action, taken by people either deliberately or accidentally that impact cybersecurity" [8]. This source of operational risk includes actions of people such as Inadvertent (mistakes, errors, omissions), deliberate (fraud, sabotage, theft, vandalism), and inaction (skills, knowledge, guidance, and availability). Systems and technology failures are referred to as issues that are abnormal or unexpected that hit technology assets [8]. This source of operational risk includes systems and technology failures such as hardware (capacity, performance, maintenance, and obsolescence), software (compatibility, configuration management, change control, security settings, coding practices, and testing), and system (design, specifications, integration, and complexity).

To contain cyber risks, the health facilities must implement an efficient risk management processes. This process must support the health facility to reach the organization's aims. Risk management is a continuous process that depends directly on the changes in the internal and external environment of the organization [5].

Using a systematic and rigorous approach, the paper aims to analyze the literature on the cyber risk in the healthcare sector to highlight the real knowledge on this topic. First, it analyzes the main bibliometric information on cyber risk to understand the main publication trend on this topic. Second, it classifies the documents identified according to the main risk dealt with. The findings illustrate that not enough studies on cyber risk in the healthcare sector. This sector needs innovative managerial solutions to face cyber-attacks, especially after the COVID-19 pandemic.

The article is organized as follows. Firstly, it introduces the materials and methods this research used to collect and analyze the literature. Secondly, it reveals and discusses the main results of this study. The last section summarizes the contributions, implications, and limitations of this research.

## 2. Materials and Methods

To realize a rigorous systematic literature review, we followed the guideline suggested by Tranfield et al. (2003). As already used by other authors [10–12], to make the literature review replicable, transparent, and scientific, we adopted the five stages suggested by Tranfield et al. (2003). Although there are many approaches to carry out a systematic literature review, we adopted the approach proposed to Tranfield et al. (2003) because it is one of the most famous approaches in the managerial literature. It has more than 6000 citations on Google Scholar, 2500 citations Web of Science, and 3000 citations Elsevier's Scopus. It is one of the most recognized, tested, and validated by the research community.

*Step 1. Planning the systematic literature review and identifying keywords*. We planned this review thanks to the collaboration of academics, practitioners, and consultants active in the field to define the keywords for the review process [13]. We created a research group useful to validate our research process. The group included professors, researchers, risk managers, insurers, doctors, and lawyers. We carried out 12 meetings to better understand this research field with this research group. Thanks to these meetings, we identified the keywords needed for the literature review. The keywords identified are "cyber", "computer security", "health", and "risk". "Cyber" includes all keywords linked to this term such as "cyber risk", "cyber-attack", cybersecurity", etc. "Health" includes keywords such as

"healthcare sector", "healthcare", "health facilities", etc. Finally, the "risk" includes all keywords such as "risk management", "risk assessment", "risk evaluation", etc.

*Step 2. Defining the criteria of document selection.* We chose peer-reviewed literature available on Elsevier's Scopus and Web of Science because it is the best scientific database in the field [14]. The criteria of document selection are described in Table 1. We restricted the search of keywords to abstract, titled, and keywords, articles, and reviews published in journals. We performed this final search on 29 June 2020.

**Table 1.** Research Criteria.

| Dataset | Elsevier's Scopus and Web of Science | | |
|---|---|---|---|
| **Time** | From the first publication (date 1992) to 2020 | | |
| **Document Type** | Article and Review | | |
| **Source Type** | Journal | | |
| **Keywords** | "Cyber" or "Computer security" | and | "Health" and "Risk" |

*Step 3. Extracting the relevant documents.* We extracted 419 documents from Scopus and Web of Science. After, we read the title of these 419 publications and selected 149 publications. Finally, we read the abstract of these 149 publications and selected 84 documents useful to reach the aim of this study (Table 2).

**Table 2.** Selection process of relevant documents.

| | |
|---|---|
| 1 | We read 419 publications' titles from Scopus and Web of Science and selected 149 publications |
| 2 | We read 149 publications' abstracts and selected 84 documents useful to the aim of the research |
| 3 | We read 84 publications to describe the main information on cyber risk in the health facilities |

*Step 4. Classifying of information.* Using a datasheet created on Excel Office, we carried out the following analyzes subdivided into two groups:

(a) Publications' trend—number of documents and citations by year, subject areas, documents by country, most keywords, and methodological information (theoretical research, e.g., literature review, descriptive; or empirical research, e.g., case study, action research);

(b) Document information—aim and risk/s dealt for each paper.

To classify the risk discussed to each paper, we used one of the most known as "*Taxonomy of operational cybersecurity risks*" proposed by Cebula and Young (2010) [8]. It will be subsequently illustrated in the paper.

*Step 5. Discussion and validity of results.* We analyzed the findings describing the main information and future opportunities on cyber risk in health facilities. We assessed the validity of the research process and the results. As said before, this review is a high-level overview of primary studies on a specific issue that identifies, selects, synthesizes, and appraises high-quality research evidence relevant to that issue [13]. It responds to a focused question that eliminated any bias. The validity of results is led to clear selection criteria, systematic search strategy, and reference list. Furthermore, to assess the validity of results we controlled the use of the criteria suggested by the literature [13]. Specifically, we controlled the following criteria useful to a rigorous systematic literature review, i.e., (a) *explicit*, a protocol describes the selection criteria; (b) *replicable*, based on transparent process; (c) *qualitative*, based on quality assess; (d) *inductive*, biases are reduced to motivations; (e) *collaborative*, a research group assessed the review process, and (f) *international*, developed through international databases to make it available to reviewers.

## 3. Results

We reviewed 84 publications on cyber risk from 1995 to 2020, describing the main: (a) publications' trend and (b) document information.

*(a) Publications' trend.* The first group of results describes the publications' trend on cyber risk in the healthcare sector. It outlines the number of documents and citations by year, subject areas, documents by country, most keywords, and methods (see Figures 1 and 2). The analysis of document numbers by year illustrates the considerable scholars' effort on the topic in the last three years. From 2017, the study of cyber risk in the healthcare sector described an increase in the number of publications, from 0 to 12 documents. However, the number of citations is still limited (see Figure 1).

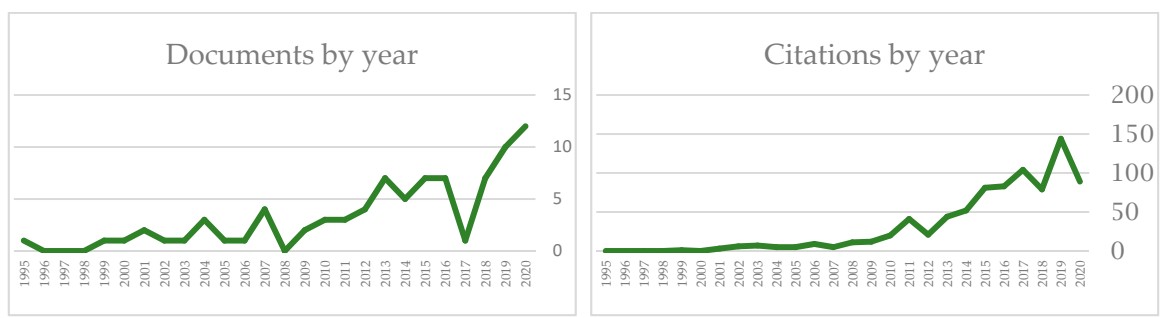

**Figure 1.** Publications' trend—documents and citations by year.

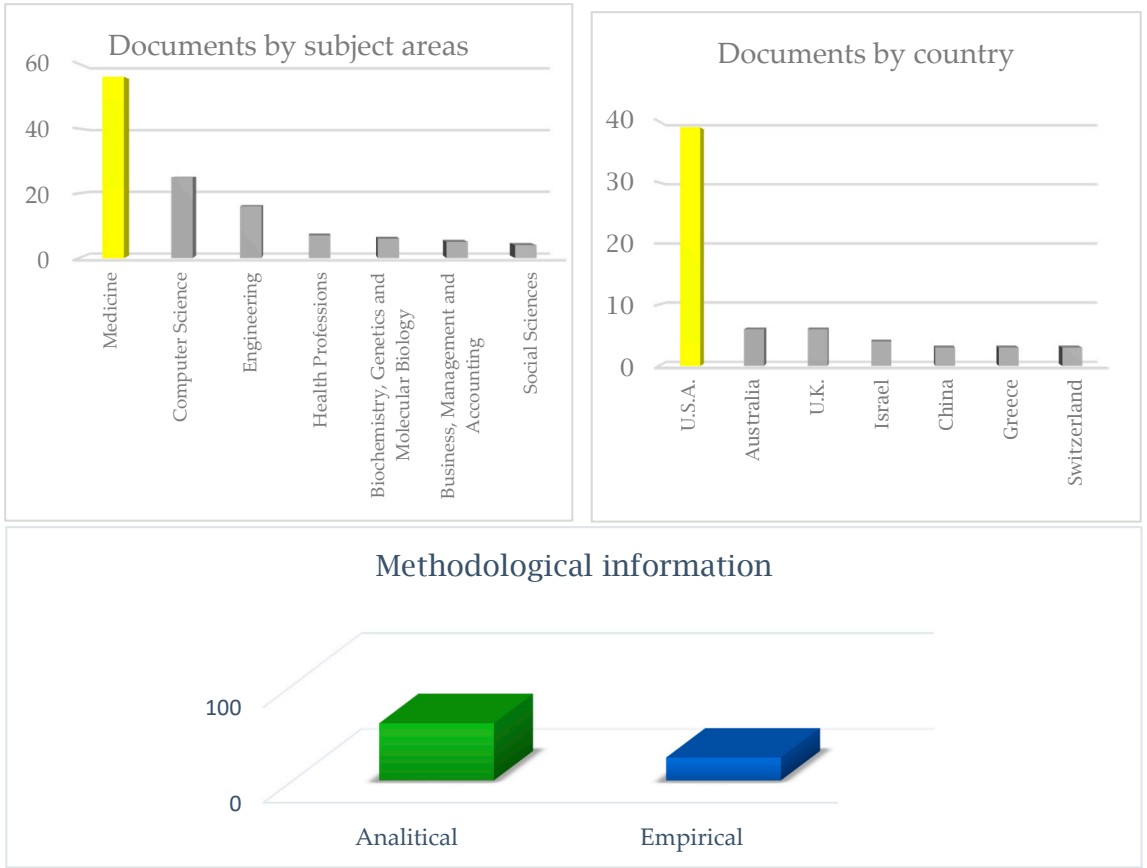

**Figure 2.** Publications' trend—subject areas, documents by country, and most methods.

The main subject areas are Medicine (40%, 56 documents), Computer Science (17.9%, 25 documents), and Engineering (11.4%, 16 documents). Not enough documents investigated other subject areas such

as Social Science (2.9%, 4 documents), Business, Management and Accounting (3.6%, 5 documents), and Mathematics (1.2%, 1 document). Another significant publications' trend described that the most prolific country was the United Stated (46%, 39 documents), which is considered the top country. The United States investigated cyber risk concerning other topics such as risk management and risk assessment. After the United States, the most prolific countries were Australia (7.1%, six documents) and the United Kingdom (7.1%, six documents). Another analysis regards the methodological information; this investigation highlights that 60 publications were based on the analytical method such as literature and descriptive reviews, whilst 24 publications were based on the empirical method such as case studies and action researches (see Figure 2).

Finally, the last publications' trend outlines the ranking of most keywords (see Table 3). It describes a clear prevalence of documents focused on computer security (68 documents) rather than cybersecurity (No. 11 documents). Furthermore, it is recognized as key topics 'risk management' and 'risk assessment'. Cyber risk and cyber-attack are not included in the ranking.

**Table 3.** Most keywords.

| Keyword | No. | Keyword | No. |
|---|---|---|---|
| Computer Security | 68 | Health Insurance | 17 |
| Human | 38 | Electronic Medical Record | 16 |
| Risk Management | 35 | Electronic Health Records | 14 |
| Risk Assessment | 32 | Medical Information System | 14 |
| Confidentiality | 30 | Electronic Health Record | 12 |
| Humans | 30 | Health Insurance Portability and Accountability Act | 12 |
| United States | 24 | Internet | 12 |
| Health Risks | 21 | Medical Informatics | 12 |
| Privacy | 21 | Patient Information | 12 |
| Health Care | 19 | Security of Data | 12 |
| Organization and Management | 19 | Cyber Security | 11 |
| Priority Journal | 18 | Review | 11 |

*(b) Document information.* The second group of analysis describes the overall aim of the publications (Appendix A) and class/subclass of risk analyzed in each paper (Table 4). In Appendix A and Table 4, we summarized the aim and the risk of each paper.

The documents analyzed deal mainly the potential of using technology for various purposes related to the healthcare sector (Appendix A). For example, telemedicine, electronic health/medical record, and mobile health application. Telemedicine is defined by the World Health Organization (p. 9) as: *"The delivery of health care services, where distance is a critical factor, by all health care professionals using information and communication technologies for the exchange of valid information for diagnosis, treatment and prevention of disease and injuries, research and evaluation, and for the continuing education of health care providers, all in the interests of advancing the health of individuals and their communities"* [15]. Telemedicine is used such as supports the diagnosed and medications prescribed via untact [16]. Electronic Health Record (EHR), or Electronic Medical Record (EMR), is recognized as the storing of patients' medical record in a different way from the traditional medical record [17]. EHR is described as the ability to store a large amount of low-cost medical records using devices and ensure better patient care [17]. Mobile health application, so-called mHealth, is the technology using to medical personnel for medical and other health-related purposes [18]. This technology allows patient care; however, it can be attacked by hackers because it has a significant financial value in terms of personal data [18].

This analysis points out great attention to subclasses and classes such as Deliberate (Actions of People), Software (System and Technology Failures), and Process controls (Failed Internal Processes) (see Table 4).

**Table 4.** Risk analyzed in each paper according to Taxonomy of Operational Cyber Security Risks [8].

| Class | Subclass | No. | References |
|---|---|---|---|
| **1 Actions of People** | 1.1 Inadvertent | 4 | [19–22] |
| | 1.2 Deliberate | 13 | [23–35] |
| | 1.3 Inaction | 3 | [36–38] |
| **2 Systems and Technology Failures** | 2.1 Hardware | 1 | [39] |
| | 2.2 Software | 42 | [16,18,37,40–67] |
| | 2.3 Systems | 1 | [68] |
| **3 Failed Internal Processes** | 3.1 Process design or execution | 1 | [69] |
| | 3.2 Process controls | 16 | [17,70–84] |
| | 3.3 Supporting processes | 1 | [85] |
| **4 External Events** | 4.1 Hazards | 0 | |
| | 4.2 Legal issues | 0 | |
| | 4.3 Business issues | 2 | [86,87] |
| | 4.4 Service dependencies | 0 | |

## 4. Discussion

This paper analyzed publications' trend and aim, and risk/s dealt for each paper. The findings of these two analyses highlight two groups of results.

The first group of results, based on the analysis of publications' trends, identifies numerous future research opportunities. Although the literature describes a considerable scholars' effort on cyber risk in the last two years, the results highlight a great need for further research. The total number of documents is not enough to answer to the cyber risk management challenge in the healthcare sector. The studies included in the Medicine area are not enough to contribute to cyber-risk management as it lacks to risk assessment based on specific needs of health facilities. This topic should be investigated by a holistic and multidisciplinary approach to respond to the management challenge. The paper highlights a poor use of risk assessment tools to analyze cyber risk. Although the results highlight the use of the most important reactive risk assessment tools in the healthcare sector such as Incident Reporting [80,81,88], Root Cause Analysis [31,69], and Failure Mode and Effect Analysis [16], it describes not enough studies useful to investigate this topic. Furthermore, the results describe the main cyber risks applied at the management of electronic medical records [33,41,62], electronic health records [26,55,71,89], telemedicine devices [16], and mobile health [38,40,62,83]. The literature analyzed calls for studies to other subject areas such as Business, Management and Accounting; Social Science; and Mathematics. Practical experience from many countries must be known to provide new theory and develop tools able to answer a current world situation. Scholars should also move research focus from computer security to cyber risk; cyber risk is referred to as the probability that a given source of cyber threat exerts a potential vulnerability and the consequent impact of this adverse event on the organization. Scholars should pay attention to cyberspace as it is the place where adverse events can occur. Cyberspace is an interactive domain composed of all digital networks used to store, modify, and communicate information. It includes all information systems used to support businesses, infrastructures, and services [7].

Highlights of the first analysis:

- The considerable scholars' effort on the topic in the last two years. However, the results describe a great need for further research. The total number of documents is not enough to answer to the cyber risk management challenge in the healthcare sector.
- The Medicine area as the most subject area. The literature calls studies to other subject areas such as Business, Management and Accounting, Social Science, and Mathematics.

- The United States as the most prolific country. This analysis outlines a gap in the study of this topic in many countries.
- The analytical method as the most research approaches utilized. The inquiry encourages empirical research to contribute to practical knowledge on this topic.
- Computer security, risk management, and risk assessment were the most often used keywords. There are not enough studies that use cyber risk such as keyword.

The second group of results, based on the analysis of aim and risk/s dealt in each paper, describes the cyber risks in the healthcare sector. It highlights the risks in the use of telemedicine, electronic medical record, and mobile health. This literature review illustrated that there are not enough studies about the cyber risk assessment. Finally, the document analyzed describes some key classes of operational cybersecurity risks such as Deliberate, Software, and Process controls. Furthermore, this analysis highlights a gap in the study of "External Events", i.e., Hazards, Legal issues, Business issues, and Service dependencies. The literature on the topic of cyber risk in the healthcare sector has been increasing in interest over the past few years. The main information technology risk for the health facilities, and generally for the Company, is the manipulation of the cyber-based system made by hackers, criminal or dishonest people to take information about people accesses in the Company system. The cyber-based system is a technology infrastructure that organization can use to simplify the work of the people. The manipulation of this system can be a risk for everyone because the attacker can obtain key company data relative to their business. It is important to protect from cyberattacks because they caused serious damage and the defense of these attackers is not an easy task.

The use of this technology supports doctors and employees in their duties through remote work for health treatment and administrative purposes [16]. However, it alerts the health facilities and its stakeholders on dangers derived from the use of these technologies [90]. This danger can derive from internal problems—e.g., the wrong design of the computer network or inefficient processes [22]—and external problems—e.g., cyber-attack by hackers [24]. These dangers are increasingly widespread and difficult to control all activities [91]. For instance, an activity with a high damage probability is telemedicine [16]. An increasingly widespread cyber risk is that of remote access to electronic medical records patients [18]. This access can be used as a fraudulent way by hackers to access information in the databases [17]. In addition to technology, hackers can also take advantage of staff loopholes and carelessness to collect data [19].

As highlighted by some papers, the solutions to improve cyber-risk management are the continuous training of employees [19], the use of performing technologies, the continuous process improvement [49], the implementation of risk management activities [48], the use of proactive and reactive risk assessment tools [52], and the stipulation of insurance policies to protect any damage to stakeholders and the health facilities [71]. The device defense systems (antimalware, security patches, and software) must be updated constantly. Furthermore, the personnel of health facilities must be periodically trained on potential new threats to protect patient health and business risk [53]. To protect business risk, the literature highlights the need to implement risk management activities. It pushes the use of proactive and reactive risk assessment tools. The implementation of these activities and tools allows to know, manage, and contain the risk effectively. Besides, the literature highlights the use of insurance policies to protect asset management from health facilities [71].

The highlights of the second analysis are as follows:

- Good knowledge of cyber risks was linked to the use of technology in the healthcare sector (e.g., telemedicine, electronic medical record, and mobile health). However, there are not holistic studies that introduce all cyber risks linked to the use of technology in the healthcare sector.
- Numerous publications related to the study of some subclasses of operational cybersecurity risks such as Deliberate, Software, and Process control topics. Furthermore, this analysis outlines a gap in the study of the class 'External Events'.

## 5. Conclusions

This paper presents a systematic literature review on cyber risk in the healthcare sector. It describes the main literature information on cyber risk. It highlights the poor attention of the scientific community on this topic, except in the United States. The studies related to the health facilities are not enough to answer healthcare needs. The literature lacks research contributions to face the cyber risk management challenge in the healthcare sector. This topic should be developed in other countries and subject areas such as Business, Management and Accounting; Social Science; and Mathematics. The results of this research highlight the need for further studies to investigate empirically the cyber risk especially connected to some classes and subclasses of operational cybersecurity risks. For instance, scholars should provide more contributions to External Events which hazards, legal issues, business issues, and service dependencies.

The implications of this research are twofold. One the one hand, it highlights knowledge of the literature on the cyber risk. On the other hand, it identifies gaps in the literature which need to be filled and, consequently, future research opportunities.

This research has a main limitation, i.e., it analyzed only the documents related to the keyword "health"; this criterion may narrow the field excessively. However, we chose this keyword strategy to understand the current situation on cyber risk in the healthcare sector, especially during the COVID-19 pandemic.

This limitation may also be the strength of this research. Thanks to this research criterion, it identifies knowledge gaps in the literature and offers future research opportunities in studying cyber risk. Firstly, scholars may investigate the literature on cyber risk in other sectors and replicate the best practices in the health facilities. Secondly, it encourages new managerial solutions derived from practical experiences of consults and practitioners.

**Author Contributions:** Introduction, A.S. and E.S.; Methodology, A.S., E.S., and A.G.; Findings A.S., A.R., and E.S.; Discussion A.S., A.R., and E.S.; Conclusions A.S., A.R., E.S., and A.G. All authors have read and agreed to the published version of the manuscript.

**Funding:** This research was funded by Sham—Societè Hospitaliere D'assurances Mutuelles—for financing the Research scholarship no. 10/2020, Department of Management, University of Turin, entitled: Identification, analysis, and mapping of risks and harmful events c/o health organizations.

**Acknowledgments:** We would thank our research group for their technical support. The group included professors, researchers, risk managers, insurers, doctors, and lawyers.

**Conflicts of Interest:** The authors declare no conflict of interest.

## Appendix A

| Publications' Aim | Subclass |
|---|---|
| to explain criminal behavior reliant on computing and the online domain with particular characteristics and motivations such as being young, male, autistic and motivated by challenge [92] | 1.2 |
| to explain like most breaches are the result of employee carelessness and/or failure to comply with information security policies and procedures, but to external hackers, too [19] | 1.1 |
| to empirically test a proposed conceptual model, using integrated concepts from the Theory of Planned Behavior, the Information Security Policy Compliance Theory, and the aggregated Revealed Causal Map of EMR Resistance [36] | 1.3 |
| to explain the key construction processes of the model which include initialization, data appending, scale expansion, data query, and verification to protect the integrity and privacy of the healthcare-related data [93] | 2.2 |
| to analyze the risks and security threats comprehensively and institute appropriate countermeasures to protect patients and improve telemedicine quality for patient safety [16] | 2.2 |
| to examine parent perspectives about electronic consultations, including perceived benefits and risks, anticipated informational needs, and preferences for parent engagement with electronic consultations [94] | 2.2 |

| | |
|---|---|
| to explain like biosecurity can be dangerous for data breaches and disruption of operations at biological facilities from cyber-attacks [88] | 2.2 |
| to explore cybersecurity aspects of microbial NGS and to discuss the motivations and objectives for such as attack, its feasibility and implications, and highlight policy considerations aimed at threat mitigation [89] | 2.2 |
| to present a risk assessment feature integrated into the Socio-Technical Risk-Adaptable Access Control model, as well as the operationalization of the related mobile health decision policies [18] | 2.2 |
| to present a deep recurrent neural network solution as a stacked long short-term memory with a pre-training as a regularization method to avoid random network initialization [95] | 2.2 |
| to explain like physical systems are influenced by dynamic and evolving technologies, environments, and attack mechanisms with rapidly changing and difficult to detect and manage the vulnerabilities [70] | 3.2 |
| to examine the potential cyber risks arising from the application of IoT devices-linked insurance [71] | 3.2 |
| to report on an internal evaluation targeting hospital staff and summarize peer-reviewed literature regarding phishing and healthcare [24] | 1.2 |
| to classifying the variety of cyber risks so that they can be addressed appropriately and can help to develop a common language for the science [91] | 2.2 |
| to present a taxonomy of ten widely-used PMDs (personal medical devices) based on the five diseases they were designed to treat and to provide a comprehensive survey that covers 17 possible attacks aimed at PMDs, as well as the attacks' building blocks [90] | 2.2 |
| to present a systematic identification and evaluation of potential privacy risks, particularly emphasizing controls and mitigation strategies to handle negative privacy impacts [40] | 2.2 |
| to propose a fog computing security and privacy protection solution and to design the security and privacy protection framework based on the fog computing to improve telehealth and telemedicine infrastructure [64] | 2.2 |
| to detail the development and execution of three novel high-fidelity clinical simulations designed to teach clinicians to recognize, treat, and prevent patient harm from vulnerable medical devices [39] | 2.1 |
| to determine whether the approach used in Australia to regulate mobile medical applications is consistent with international standards and is suitable to address the unique challenges of these technologies [41] | 2.2 |
| to define several potential cybersecurity weaknesses in today's pathogen genome databases to raise awareness [42] | 2.2 |
| to propose a novel maturity model for health-care cloud security, which focuses on assessing cyber security in cloud-based health-care environments by incorporating the sub-domains of health-care cyber security practices and introducing health-care-specific cyber security metrics [72] | 3.2 |
| to use innovative technology in healthcare to treat, diagnose and monitor patients [43] | 2.2 |
| to investigate medical information security to gain a better understanding of trends in research related to medical information security [96] | 1.2 |
| to present a novel approach, called BotDet, for botnet Command and Control traffic detection to defend against malware attacks in critical ultrastructure systems [44] | 2.2 |
| to develop a model of factors associated with healthcare data breaches. Variables were operationalized as the healthcare facilities' level of exposure, level of security, and organizational factors [45] | 2.2 |
| to record public and physicians' awareness, expectations for, and ethical concerns about the use of EHRs [46] | 2.2 |
| to provide a minimal level of cybersecurity, but there are deficiencies in the standard and identifies the important aspects of cybersecurity that could be improved [73] | 3.2 |
| to exploit of cybersecurity vulnerabilities can affect fielded medical devices today. Indeed, unmitigated cybersecurity vulnerabilities have already led to medical devices being infected and disabled by malware [74] | 3.2 |
| to develop an enterprise risk inventory for healthcare organizations to create a common understanding of how each type of risk impacts a healthcare organization [86] | 4.3 |
| to establish that stakeholders have a shared responsibility to address cybersecurity threats that can affect such devices [47] | 2.2 |
| to explain like hackers attack healthcare aren't after credit card numbers; they're looking for data-rich electronic health records [26] | 1.2 |

| | |
|---|---|
| to explain the heightened interest and increased spending on health IT security [27] | 1.2 |
| to describe the underlying causes of some of the largest health care data breaches of the past several years and provide practical advice on how future data breaches could be prevented [28] | 1.2 |
| to describe health care breaches of protected information, analyze the hazards and vulnerabilities of reported breach cases, and prescribe best practices of managing risk through security controls and countermeasures [48] | 2.2 |
| to explain a new health record storage architecture, the personal grid eliminates this risk by separately storing and encrypting each person's record [68] | 2.3 |
| to explain like new vulnerabilities can emerge from the malicious behavior of threat actors and these attacks can be sudden and unexpected [49] | 2.2 |
| to explain like organizations must look at different approaches to data protection [87] | 4.3 |
| to present several security attacks on Lu et al.'s protocol such as identity trace attack, new smart card issue attack, patient impersonation attack and medical server impersonation attack [29] | 1.2 |
| to monitor the high-risk patients and to protect the patient's data from intruders at anytime and anywhere through android APP [30] | 1.2 |
| to explain like medical devices can be attacked from hackers and the role of companies to create a security system [50] | 2.2 |
| to describe a methodical process to ensure medical device cybersecurity at a 400-bed tertiary care medical center [51] | 2.2 |
| to explain the cyber risk management for the healthcare industry [52] | 2.2 |
| to evaluate whether potential users in healthcare organizations can exploit the GST technique to share lessons learned from security incidents [75] | 3.2 |
| to explain like cybersecurity protection is not just a technical issue; it is a richer and more intricate problem to solve [76] | 3.2 |
| to re-examine and analyze the causal factors behind healthcare data breaches, using the Swiss Cheese Model to shed light on the technical, organizational, and human factors of these breaches [31] | 1.2 |
| to include the effects of medical identity fraud on patient compliance, brand, and profitability [32] | 1.2 |
| to explore the importance of medical device cybersecurity and the consequences of security breaches [53] | 2.2 |
| to explain like preventing data breaches has become more complex, and at the same time, the fines being levied against health care organizations for violating the Health Insurance Portability and Accountability Act Privacy and Security Rules are becoming larger [54] | 2.2 |
| to propose a framework that includes the most important security processes regarding cloud computing in the health care sector [77] | 3.2 |
| to suggest that cyber threats are increasing and that much of the U.S. healthcare system is ill-equipped to deal with them [33] | 1.2 |
| to discuss the actions taken by standards bodies, such as the Association for the Advancement of Medical Instrumentation, to improve medical device cybersecurity [55] | 2.2 |
| to identify and sketch the policy implications of using HSNS and how policymakers and stakeholders should elaborate upon them to protect the privacy of online health data [67] | 2.2 |
| to risk assessment of privacy and security aspects has been performed, to reveal actual risks and to ensure adequate information security in this technical platform [56] | 2.2 |
| to build on a novel combination of virtualization and data leakage protection and can be combined with other protection methodologies and scaled to production level [57] | 2.2 |
| to explain what people can do if the protected information is breached [58] | 2.2 |
| to focus on protecting all ePHI stored in and transmitted via smartphones. This includes a cryptographic scheme required to address the problem [78] | 3.2 |
| to describe why incorporating an understanding of human behavior into cybersecurity products and processes can lead to more effective technology [59] | 2.2 |
| to address cyber threats, governments, industry, and consumers should support collective cyber defenses modeled on efforts to address human illnesses [60] | 2.2 |

| | |
|---|---|
| to present a detailed public health framework-including descriptions of public health threats encountered and interventions used-and develop parallels between public health and cybersecurity threats and interventions [79] | 3.2 |
| to explain like a threat modeling methodology, known as attack tree, is employed to analyze attacks affecting EHR systems [17] | 3.2 |
| to not only develop policies and procedures to prevent, detect, contain, and correct security violations, but should make sure that such policies and procedures are implemented in their everyday operations [20] | 1.1 |
| to address the problem of improper use of health data and introduce a methodology that protects medical records from unauthorized access, leaving the patient the choice to decide which people are authorized to use his data [34] | 1.2 |
| to emphasis on security issues, which can arise inside a virtual healthcare community and relate to the communication and storage of data [21] | 1.1 |
| to provide an overview of the current methodologies used to ensure data security, and a description of one successful approach to balancing access and privacy [37] | 1.3 |
| to examine the security issues for the implementation of e-healthcare using currently available healthcare standards and proposes solutions and recommendations to secure the future of e-healthcare [35] | 1.2 |
| to present the essential requirements, critical architectures, and policies for system security of regional collaborative medical platforms [61] | 2.2 |
| to analyze clinicians' health information system privacy and security experiences in the practice context [62] | 2.2 |
| to preserve the privacy and security of patients' portable medical records in portable storage media to avoid any inappropriate or unintentional disclosure [63] | 2.2 |
| to propose MedIMob for a secure enterprise IM service for use in healthcare. MedIMob supports IM clients on mobile devices in addition to desktop-based clients [97] | 2.2 |
| to explain like the consequences of a cyber-attack or privacy breach could be operationally and financially catastrophic, so an HCO's move toward an enterprise-wide approach at identifying and minimizing risk, cyber and privacy liability should be on the radar screen for risk managers and leadership [98] | 2.2 |
| to develop guidelines for computer security in general practice based on a literature review, an analysis of available information on current practice and a series of key stakeholder interviews [99] | 2.2 |
| to develop a model-based approach towards end-to-end security which is defined as continuous security from point of origin to point of destination in a communication process [80] | 3.2 |
| to guide the security essentials necessary to promote best practice for information security [81] | 3.2 |
| to explain that the system addressed threats and vulnerabilities in the privacy and security of protected health information [85] | 3.3 |
| to explain like the software program began an insidious assault on the hospital's network, seeking out and copying files from every hard drive it could find [22] | 1.1 |
| to explain like who get involved in security compliance can be unique and valuable assets to their organizations and to patient privacy [38] | 1.3 |
| to describe information security design, implementation, management, and auditing inside a multi-specialty provincial Italian hospital [100] | 3.2 |
| to explain like information systems using public or private networks become vulnerable to outside attacks every time new servers are added or firewalls are updated [101] | 2.2 |
| to explain like information technology is a key component in both defending against and aiding terrorism threats and other forms of terrorism, cybersecurity - national (and global) critical information infrastructure protection [66] | 2.2 |
| to explain like organizations must embark on an arduous journey to identify their vulnerabilities and come up with strategies to plug their security holes. To do so, they must conduct a gap analysis to determine those vulnerabilities and a risk assessment to set a policy framework [83] | 3.2 |
| to explain like healthcare risk managers should be aware of their organizations' electronic activities, the new risks brought about by these activities and alternative measures that can be taken to reduce or transfer the risks [84] | 3.2 |

| to present the results of a risk analysis, based on the CRAMM methodology, for a healthcare organization offering a patient home-monitoring service through the transmission of vital signs, focusing on the identified security needs and the proposed countermeasures [65] | 2.2 |
| --- | --- |
| to give an overview of current trends in the security aspects of health-care information systems [102] | 2.2 |
| to examine the nature of security in the context of health care and explores the importance of the identification of risk [69] | 3.1 |

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
