# Peer review of "Cyber Risk in Health Facilities: A Systematic Literature Review"

_sustainability, doi:10.3390/su12177002_

Round 1

Reviewer 1 Report

As well stated by the authors, the paper provides a systematic literature review with focus on statistical information (how often the topic is covered, number of documents by subject areas and countries, etc.), aims (given on roughly 4 1/2 pages in Annex A) and use of key words in literature rather than on technical results. As a way out I do see two options, either to extend publications' aim by publications' results with regard to cyber risks in the Annex and in the text, respectively, or to modify the title, e.g., to "Coverage of cyber risk in health facilities in literature".

In my view some limitations of the review are not sufficiently well reflected, e.g., the limitation on the English language and the (resulting (?)) highligh-ted attention in the US while notably German, French and researchers from other countries in this field may still publish in their own language.

Finally, please check information on read and selected publications (line 91-96), separate captions from running text (line 94/98) and further streamline the article by avoiding duplications, e.g., definitions and explanations on sources of operational risks are given twice - in the introduction and results (lines 169-183). A few words on risks due to manipulation of cyber-based system deem appropriate. 

Author Response

Dear Reviewer,

Thank you so much for your positive feedback. We are pleased to submit a new version of the manuscript. We have carefully taken into account your comments and suggestions, which provided us with very helpful and constructive feedback.

We are very grateful for your contribution to improving our study. If you find that the paper requires further improvement or revision, we will be glad to accommodate such changes.

Below, we respond to each of your suggestions (answers are reported in bold format)

As well stated by the authors, the paper provides a systematic literature review with focus on statistical information (how often the topic is covered, number of documents by subject areas and countries, etc.), aims (given on roughly 4 1/2 pages in Annex A) and use of key words in literature rather than on technical results. As a way out I do see two options, either to extend publications' aim by publications' results with regard to cyber risks in the Annex and in the text, respectively, or to modify the title, e.g., to "Coverage of cyber risk in health facilities in literature"

Thank you for this suggestion. We agree with your comment and chose to modify the title as follows: "Cyber risk in health facilities: A systematic literature review".

In my view some limitations of the review are not sufficiently well reflected, e.g., the limitation on the English language and the (resulting (?)) highlighted attention in the US while notably German, French and researchers from other countries in this field may still publish in their own language.

Thank you for this comment. We deleted the criterion of "English language". Then, we checked the documents published in other languages and found other 12 documents published in German (9), Chinese (1), French (1), and Italian (1). Now the documents moved from 401 to 413. Thanks to the translation support of some colleagues, we analyzed these documents. However, we did not find significant information for our research. Probably, there are not many documents because we used the peer-reviewed literature available on Elsevier’s Scopus and Web of Science, the main scientific worldwide databases which publish mainly in English.

Finally, please check information on read and selected publications (line 91-96), separate captions from running text (line 94/98) and further streamline the article by avoiding duplications, e.g., definitions and explanations on sources of operational risks are given twice - in the introduction and results (lines 169-183). A few words on risks due to manipulation of cyber-based system deem appropriate. 

Thank you for these suggestions. 

  1. We edited the information on reading and selected publications (line 91-96), separate captions from running text (line 94/98). We wrote the following sentence:

    "Step 3. Extracting the relevant documents. We extracted 401 documents from Scopus & Web of Science. After we read the title of these 401 publications and selected 149 publications. Finally, we read the abstract of these 149 publications and selected 84 documents useful to reach the aim of this study (Table 2)."

  2. We edited/deleted duplications (e.g. lines 169-183)
  3. We included few words on the risk due to the manipulation of the cyber-based system in Section 4 (discussion).

As you notice, we addressed all your comments. We would like to take this opportunity to express our sincere who identified areas of our manuscript that needed improvements. The manuscript has certainly benefited from these insightful revision suggestions.

Thank you again for your useful support.

Best regards,

The Authors

Reviewer 2 Report

The paper is very interesting, and formally it is a very good work. However, the scientific content, I think, is low. Sections 2 and 3 are very interesting and provide a quantitative analysis of relevant papers on health cybersecurity. Nevertheless, Section 4 I think is too short and poor. I greatly recommend to include more details and additional discussions at this point. A deeper qualitative analysis is necessary.

For example, two different groups of works ar identified but, are they homogenous? can you identify subgroups? Is there any relation between, for example, the publication year and the content? Probably the view depends on the authors' expertise? What are the validity threats to your methodology and experiment? 

Many questions are still unanswered in this paper, and I think they should be addressed before acceptance

Author Response

Dear Reviewer,

Thank you for your positive feedback.

We are pleased to submit a new version of the manuscript following your suggestions. We have carefully taken into account your comments, which provided us constructive feedback.

We are very grateful for your contribution to improving our study. If you find that the paper requires further improvement or revision, we will be glad to accommodate such changes.

Below, we respond to each of your suggestions (answer is reported in bold format)

The paper is very interesting, and formally it is a very good work. However, the scientific content, I think, is low. Sections 2 and 3 are very interesting and provide a quantitative analysis of relevant papers on health cybersecurity. Nevertheless, Section 4 I think is too short and poor. I greatly recommend to include more details and additional discussions at this point. A deeper qualitative analysis is necessary. For example, two different groups of works ar identified but, are they homogenous? can you identify subgroups? Is there any relation between, for example, the publication year and the content? Probably the view depends on the authors' expertise? What are the validity threats to your methodology and experiment? 

Thank you for this suggestion. In accordance with your comments, we improved the discussion of the results. In specific, we integrated Section 4 included more details and additional discussions at this point though a deeper qualitative analysis. Moreover, we moved the highlights of this research from Section 3 to Section 4 in order to improve the paper design and provide more information in the Discussion section. Furthermore, we explained better two different groups identified during the analysis, but you did not find significant relations between the publication year / content and the authors' expertise. The authors' expertise follows the main subject area (Figure 2). Finally, we included the validity threats to the methodology. In specific, we included this sentence:

"According to Tranfield et al. (2003), we assessed the validity of the research process and the results. As said before, this review is high-level overview of primary studies on a specific issue that identifies, selects, synthesizes, and appraises high-quality research evidence relevant to that issue. It responds to a focused question that eliminated any bias. The validity of results is led to clear selection criteria, systematic search strategy, and reference list. Furthermore, to assess the validity of results we controlled the use of the criteria suggested by Tranfield et al. (2003). In specific, we controlled the following criteria useful to a rigorous systematic literature review, i.e. a) explicit, a protocol describes the selection criteria; b) replicable, based on transparent process; c) qualitative, based on quality assess; d) inductive, biases are reduced to motivations; e) collaborative, a research group assessed the review process, and f) international, developed through international databases to make it available to reviewers".

We would like to take this opportunity to express our sincere who identified areas of our manuscript that needed improvements. The manuscript has certainly benefited from these insightful revision suggestions.

Thank you for your support.

Best regards,

The Authors

Reviewer 3 Report

This work aims to analyze the literature on the cyber risk in the healthcare sector to understand the impact of COVID-19. 

This work has many issues.

- The work is only 7 pages, as a survey, it is not accepted.

- It claims to do a systematic literature review based on Tranfield et al. (2003), but the reason is unclear. In the literature, there are many other approaches can be used for such purpose.

- It fails to analyze the state-of-the-art, and the term of cyber risk is very general in this work. There is no contribution here.

- The results only show the published documents, without indicating the details. It is not helpful for the research community.

Author Response

Dear Reviewer,

Thank you for your comment. We are pleased to submit a new version of the manuscript. We have carefully taken into account reviewers' comments and suggestions, which provided us with very helpful and constructive feedback.

Below, we respond to each of your suggestions (answers are reported in bold format).

This work aims to analyze the literature on cyber risk in the healthcare sector to understand the impact of COVID-19. 

Thank you for this comment. The paper aims to analyze the literature on cyber risk in the healthcare sector to understand the real knowledge on this topic. We chose to analyze this topic because the World Health Organization has recently detected a dramatic increase in the number of cyber-attacks during COVID-19. 

The work is only 7 pages, as a survey, it is not accepted.

Thank you for your comments. Generally, the goal of a systematic literature review is to summarize the knowledge on a specific issue, i.e. what science knows and what science does not know (Tranfield et al. 2003). This research would be a systematic literature review and according to this premise, we looked to describe scientifically this knowledge.  In our opinion, this paper may be considered a springboard for further research of this as yet little explored topic. It would give "a first contribution" in order to highlight the state of art of cyber risk in health facilities.

It claims to do a systematic literature review based on Tranfield et al. (2003), but the reason is unclear. In the literature, there are many other approaches can be used for such purpose.

Thank you for your comment. We adopted one of the most famous approaches in the literature (Tranfiled et al. 2003). It obtained more than 6,100 citations on Google Scholar, 2,500 citations Web of Science, and 3,050 citations on Elsevier's Scopus. We know, there are many other approaches, however, it is one of the most recognized, tested, and validated to the research community. This is the reason for the use of this approach. We included the following sentence:

"Although there are many approaches to carry out a systematic literature review, we adopted the approach proposed to Tranfield et al. (2003) because it is one of the most famous approaches in the managerial literature. It has more than 6,000 citations on Google Scholar, 2,500 citations Web of Science, and 3,000 citations on Elsevier's Scopus. It is one of the most recognized, tested, and validated to the research community".

It fails to analyze the state-of-the-art, and the term of cyber risk is very general in this work. There is no contribution here.

Thank you for your comment. We present more definitions on cyber risk provided to numerous authors. We are sure, we do not present all the definitions; it would be impossible to identify all the definitions because there are many different views/areas/subjects that investigate this concept. However, few studies introduce a taxonomy of the operational cyber risks. Thus, we adopted one of the most recognized research (Cebula and Young, 2010) in order to define our topic. In specific, we wrote these sentences:

"Cyber risk, called also information technology risk, is the new management challenge of the third millennium; it affects the information and technology assets of organizations [2]. Scholars definite cyber risk in different ways: “the risk involved with a malicious electronic event that disrupts business and monetary loss” [3,4], “the risk is an inherent part of a business and public life” [5], or “the risk failing information systems” [6]. The term “cyber” is referred to the cyberspace, an interactive domain composed of all digital networks used to store, modify, and communicate information. It includes all information systems used to support businesses, infrastructures, and services [7]. We here define cyber risk as “operational risks to information and technology assets that have consequences affecting the confidentiality, availability, or integrity of information or information systems” [8]."

The results only show the published documents, without indicating the details. It is not helpful for the research community.

Thank you for your comment. The results describe the state of art in cyber risk in health facilities. In detail, they included the publications’ trend (number of documents and citations by year, subject areas, documents by country, most keywords, methodological information) and the document information (aim and risk/s dealt for each paper). These data should be useful to the research community to understand the real knowledge on this topic. We are sure, this paper is a springboard for further research; however, thank these data with their respective details, also illustrated by Figure 1, Figure 2, Table 3, Table 4, and Appendix A, this research should give a little contribution to the research community. 

We hope that our answers can improve your first impression. If you find that the paper requires further improvement or revision, we will be glad to accommodate such changes.

Thank you so much for your revision.

Best regards,

The Authors.

Round 2

Reviewer 1 Report

Thank you for your rebuttals and changes made. I am pleased that all my comments have been sufficiently addressed.

Reviewer 2 Report

In my opinion, the authors have addressed all my concerns and previous comments

Reviewer 3 Report

This work provides some feedback and revision, but as a review (a systematic literature review), the current form has limited contribution to the research community.

The work needs to analyze the types of cyber risks in healthcare, and systematically classify the current work into each type.